# Therapeutic miR-21 Silencing Reduces Cardiac Fibrosis and Modulates Inflammatory Response in Chronic Chagas Disease

**DOI:** 10.3390/ijms22073307

**Published:** 2021-03-24

**Authors:** Carolina Kymie Vasques Nonaka, Gabriela Louise Sampaio, Luciana de Aragão França, Bruno Raphael Cavalcante, Katia Nunes Silva, Ricardo Khouri, Felipe Guimarães Torres, Cassio Santana Meira, Emanuelle de Souza Santos, Carolina Thé Macedo, Bruno Diaz Paredes, Vinicius Pinto Costa Rocha, Silvia Regina Rogatto, Ricardo Ribeiro dos Santos, Bruno Solano de Freitas Souza, Milena Botelho Pereira Soares

**Affiliations:** 1Center for Biotechnology and Cell Therapy, São Rafael Hospital, Salvador 41253-190, Brazil; carolina.nonaka@hsr.com.br (C.K.V.N.); luaragao@gmail.com (L.d.A.F); katia.nsilva@hsr.com.br (K.N.S.); brunoparedes@gmail.com (B.D.P); brunosolanosouza@gmail.com (B.S.d.F.S.); 2Gonçalo Moniz Institute, Oswaldo Cruz Foundation (FIOCRUZ), Salvador 40296-710, Brazil; gabrielalouise.sampaio@gmail.com (G.L.S.); brunorrcavalcante@gmail.com (B.R.C); ricardo_khouri@hotmail.com (R.K.); fgtorres18@gmail.com (F.G.T); calcio0303@hotmail.com (C.S.M); souza.emanuelle@hotmail.com (E.d.S.S); carolthemacedo@gmail.com (C.T.M.); viny_rocha@hotmail.com (V.P.C.R); ricardoribeiro1941@gmail.com (R.R.d.S.); 3D’Or Institute for Research and Education (IDOR), Rio de Janeiro 20000-000, Brazil; 4Senai Institute of Innovation in Advanced Health Systems, SENAI CIMATEC, Salvador 41253-190, Brazil; 5Department of Cardiology, São Rafael Hospital, Salvador 41253-190, Brazil; 6Department of Clinical Genetics, University Hospital of Southern Denmark-Vejle, 7100 Vejle, Denmark; silvia.rogatto2@gmail.com

**Keywords:** microRNA, therapeutic target, Chagas disease, cardiomyopathy, fibrosis, miR-21

## Abstract

Chagas disease, caused by the parasite *Trypanosoma cruzi* (*T. cruzi*), remains a serious public health problem for which there is no effective treatment in the chronic stage. Intense cardiac fibrosis and inflammation are hallmarks of chronic Chagas disease cardiomyopathy (CCC). Previously, we identified upregulation of circulating and cardiac miR-21, a pro-fibrotic microRNA (miRNA), in subjects with CCC. Here, we explored the potential role of miR-21 as a therapeutic target in a model of chronic Chagas disease. PCR array-based 88 microRNA screening was performed in heart samples obtained from C57Bl/6 mice chronically infected with *T. cruzi* and serum samples collected from CCC patients. MiR-21 was found upregulated in both human and mouse samples, which was corroborated by an in silico analysis of miRNA-mRNA target prediction. In vitro miR-21 functional assays (gain-and loss-of-function) were performed in cardiac fibroblasts, showing upregulation of miR-21 and collagen expression upon transforming growth factor beta 1 (TGFβ1) and *T. cruzi* stimulation, while miR-21 blockage reduced collagen expression. Finally, treatment of *T. cruzi*-infected mice with locked nucleic acid (LNA)-anti-miR-21 inhibitor promoted a significant reduction in cardiac fibrosis. Our data suggest that miR-21 is a mediator involved in the pathogenesis of cardiac fibrosis and indicates the pharmacological silencing of miR-21 as a potential therapeutic approach for CCC.

## 1. Introduction

Chagas disease, caused by the protozoan parasite *Trypanosoma cruzi*, is among the most frequent parasitic diseases globally, with an estimated eight million infected people and 10,000 deaths annually worldwide [1]. The most severe clinical manifestation of the disease is the chronic cardiac form [2], which affects approximately 25% of the infected patients [3]. Although the mechanisms of pathogenesis are not fully understood, intense cardiac inflammation and fibrosis are hallmarks of chronic Chagas cardiomyopathy [4]. Benznidazole and nifurtimox, the two antiparasitic drugs available, have proven efficacy only in treating acute Chagas disease, but are ineffective in the chronic phase, when most patients are diagnosed [4,5]. A recent study showed that benznidazole treatment could not improve the cardiac function in patients with chronic Chagas cardiomyopathy [3].

MiRNAs are small non-coding RNAs that regulate gene expression at the post-transcriptional level [6] by regulating mRNA stability and, consequently, inhibiting translation [7]. Deregulation of miRNAs expression is involved in the pathogenesis of cardiovascular diseases [8]. Small complementary miRNA sequences can be used to inhibit miRNA activity, with promising potential for applications in clinical practice. These data suggest that miRNA target inhibition could provide significant therapeutic benefits in different disease settings, such as in cardiovascular diseases. Previous studies have identified miR-21 as an important regulator of several processes related to the pathogenesis of cardiovascular diseases, including vascular smooth muscle cell proliferation/apoptosis, cardiac cell growth/death, and cardiac fibroblast functions [8,9]. Moreover, a number of studies have shown altered miR-21 expression levels in cardiac hypertrophy and heart failure, revealing its involvement during the cardiac remodeling process [10,11,12].

Although the miRNA profile has been described in Chagas disease [13,14,15], the role of specific miRNAs and their use as potential therapeutic targets in chronic Chagas cardiomyopathy have not been explored so far. In the present study, we performed a combined PCR array and in silico analysis, and selected miR-21 as a candidate therapeutic target for Chagas disease. We evaluated the effects of blocking miR-21 in vitro and in vivo in a mouse model of Chagas cardiomyopathy.

## 2. Results

### 2.1. Screening of miRNAs in Chronic Chagas Cardiomyopathy by PCR Array and in Silico Analyses

To evaluate and select miRNAs with altered expression in Chagas disease and to identify potential therapeutic targets, we combined Polymerase Chain Reaction (PCR) array and in silico analysis. First, we evaluated the cardiac expression of miRNAs in mice chronically infected with *T. cruzi* by performing a PCR array composed of 88 miRNAs previously reported as modulated in the context of cardiovascular diseases and cardiac dysfunction. The comparison of chronically infected mice with naïve controls showed 15 upregulated miRNAs and 30 downregulated miRNAs (Figure 1A,B). To gain translational validity, these findings were compared with the results of a PCR array screening of 88 miRNAs in serum samples obtained from subjects with chronic Chagas disease cardiomyopathy and healthy controls (Figure 1C–E). Three (miR-21-5p, miR-29b-3p, and miR-29c-3p) of the five common miRNAs in the two arrays (Venn diagram) were found upregulated in both human and mouse samples (Figure 1D).

All of the miRNAs screened by PCR array were evaluated by in silico analysis using public datasets from Chagas disease studies. Five transcriptome datasets were found available in GEO (Gene Expression Omnibus) (Figure 2A), comprising in vitro (*n* = 2), in vivo (*n* = 2) studies of *T. cruzi* infection, and a study with cardiac samples from patients with Chagas cardiomyopathy (*n* = 1). Interestingly, miR-21 was the only miRNA with altered expression in all datasets, showing consistent upregulation (Figure 2A).

Next, enrichment analysis using Ingenuity Pathways Analysis (IPA) software was performed, applying the module miRNA-mRNA target from TargetScan to identify target genes in the datasets. IPA upstream regulator enrichment meta-analysis confirmed miR-21 upregulation in all datasets (*p* < 0.05), while TargetScanHuman tool (http://www.targetscan.org, accessed on 7 October 2018) identified miR-21-5p targets (Appendix A). For the next experiments, we selected two miR-21 targets predicted by the TargetScanHuman list: SPRY1 (sprouty homolog 1) and STAT3 (signal transducer and activator of transcription 3) (Figure 2B), due to the strong association as regulators of fibrosis pathways [12,13,14].

### 2.2. MiR-21 Is Overexpressed after T. cruzi Infection in Mouse Cardiac Fibroblasts

Cardiac fibrosis can be induced by tissue injury, persistent inflammation, and infection. *T. cruzi* infection activates cardiac fibroblasts, increasing the production of extracellular matrix proteins [16]. We evaluated the involvement of miR-21 in *T. cruzi* infection in mouse cardiac fibroblasts (MCF). The expression of miR-21 in MCF was increased 24 h after *T. cruzi* infection (Figure 3A). MiR-21 target genes Spry1, Cadm1 (cell adhesion molecule 1), and Stat3 were evaluated, and a significant reduction in Cadm1 mRNA was observed (Figure 3B). Tgfβ1 (profibrogenic factor) expression was also increased after *T. cruzi* infection in MCF (Figure 3B). These cells showed high immunoexpression of alpha-SMA and vimentin, compatible with a myofibroblast phenotype (Figure 3C).

### 2.3. MiR-21 Participates in the TGFβ1-Induced Collagen Pathway

To investigate the role of miR-21 in the pro-fibrotic response induced by TGF-β1 via Spry1 or Cadm1/Stat3, fibroblast cultures were stimulated with 2 ng/mL TGF-β1 for 24 h (Figure 4A). Increased miR-21 expression was significantly found in MCF after TGF-β1 stimulus (Figure 4B). Increased expression of collagen genes (Col1a1 and Col1a2) along with decreased Spry1 and Cadm1 expression were detected upon TGF-β1 stimulation (Figure 4B). Similar levels of Stat3 mRNA were detected in the same conditions.

The involvement of miR-21 in the upregulation of collagen genes induced by TGF-β1 was evaluated by performing in vitro miR-21 blockage using a synthetic LNA miRNA inhibitor. First, we confirmed the efficiency of knockdown induced by two concentrations of LNA-anti-miR-21 (10 and 25 nM) in MCF (Figure 4C). Then, we evaluated the effects of miR-21 knockdown in MCF stimulated with TGF-β1 and found that the expression of collagen genes (Col1a1 and Col1a2) were blocked in the presence of LNA-anti-miR-21. Furthermore, we found that miR-21 targets Spry1, Cadm1 and Stat3 were upregulated in MCFs upon miR-21 knockdown (Figure 4C).

Next, we evaluated whether miR-21 also plays a pro-fibrogenic role in human cardiac fibroblasts (HCF) by performing gain and loss of function studies (Figure 5A). We found that miR-21 overexpression by plasmid transfection in HCF was associated with increased collagen gene expression (*COL1A1* and *COL1A2*), reduced *SPRY1*, increased *STAT3* and a tendency to decrease *CADM1* (Figure 5B). Similarly, the cells showed high expression of alpha-SMA and vimentin (Figure 3C). The incubation with TGF-β1 increased miR-21, *COL1A1,* and *COL1A2* expression levels after 24 h, while decreasing miR-21 targets *SPRY1*, *CADM1,* and increasing *STAT3* (Appendix A). The opposite pattern of expression was observed after TGF-β1 stimulation in the presence of LNA miR-21 inhibitor without cell cytotoxic effect (Figure 5C and Appendix A). Moreover, miR-21-transfected cells showed an increased proliferation rate, as evidenced by the total number of cells and the percentage of cells found with high DNA (G2-M) content and in the S phase (EDU^+^) of the cell cycle, which were higher than control (Appendix A).

We also investigated the effect of LNA-anti-miR-21 on macrophages compared with benznidazole and scramble controls. We did not observe a reduction of *T. cruzi* infection but a decreased Interferon gamma expression and increased Arginine-1 and Interleukin-10 (Appendix A).

### 2.4. Treatment with Anti-miR-21 Induces Immunomodulatory and Anti-Fibrotic Effects in T. cruzi-Infected Mice

After showing the role of miR-21 in *T. cruzi* infection and fibrotic pathways in experiments with mouse and human cells, we addressed the role of miR-21 in the fibrogenesis during chronic Chagas cardiomyopathy by performing in vivo treatment with LNA-anti-miR-21 in mice during the chronic phase of *T. cruzi* infection (Figure 6A). The effects of LNA-anti-miR-21 were investigated using sections of mouse hearts stained with picrosirius red for fibrosis evaluation. Morphometric analysis was performed one month after the beginning of the treatment (Figure 6B–E). Administration of LNA-anti-miR-21, but not of scrambled control, caused a significant reduction in the percentage of fibrosis in the hearts of *T. cruzi*-infected mice when compared to saline-treated chagasic mice (Figure 6F). LNA-anti-miR-21 significantly reduced the cardiac expression of miR-21 (Figure 6G), and the therapeutic scheme was not associated with alterations in body weight or other signs of toxicity (Appendix A). Interestingly, a significant reduction of TNF-α (tumor necrosis factor alpha) serum levels was observed in mice treated with LNA-miR-21 inhibitor, compared to infected controls treated with saline or LNA scrambled (Figure 6H).

Anti-miR-21-treated mice presented a significant reduction of cardiac *Ptprc* (protein tyrosine phosphatase receptor type C, CD45) and *Ifng* (interferon Gamma) expression (*p* < 0.05), and a tendency to reduce the expression levels of *Tgfb1* and *Ctgf* (connective tissue growth factor) (Figure 7). MiR-21 target genes *Spry1* and *Cadm1*, however, were reduced in both LNA-miR-21 inhibitor and scrambled-treated groups (Figure 7). Interestingly, *Stat3* was upregulated in the miR-21 inhibitor-treated group.

Finally, we evaluated the expression levels of genes related to the production and degradation of the extracellular matrix. The expression of collagen genes *Col1a1* and *Col1a2* were not significantly altered when compared to scrambled controls. Metalloproteinase inhibitor 1 precursor (*Timp1)* expression was upregulated in infected mice, regardless of the treatment received, when compared to naïve controls. Matrix metalloproteinase 9 (*Mmp9*) mRNA expression was similar among the groups. In contrast, a statistically significant increase in matrix metalloproteinase 2 (*Mmp2)* expression was found in anti-miR-21 mice when compared to the other groups (Figure 7).

## 3. Discussion

In the present study, we used a combined approach to screen miRNAs with altered expression in mouse and human samples, revealing the miR-21 as a candidate therapeutic target for Chagas cardiomyopathy. The association of miR-21 and collagen production pathways was demonstrated in cultured cardiac fibroblasts, with tissue fibrosis implications. Moreover, we reported that miR-21 blockage by treatment with LNA-anti-miR-21 in the mouse model of chronic Chagas cardiomyopathy was associated with the reduction of inflammation and fibrosis, two hallmarks of the disease.

Other studies explored miRNAs role in Chagas disease [13,14,15,16] and reported the upregulation of miR-21 in mouse cardiac tissue during the acute phase of *T. cruzi* infection [14]. Herein, we showed that increased miR-21 expression is also observed during the chronic phase of *T. cruzi* infection in mice. Previous studies have shown upregulation of miR-21 in other cardiovascular diseases, suggesting its potential as a therapeutic target [17,18,19]. Targeting miR-21 using a chemically modified antagomir reduced cardiac fibrosis, cardiomyocyte size, and heart weight in a cardiac hypertrophy experimental model [20,21]. A recent study also demonstrated that antimiR-21 prevents myocardial dysfunction in a pig model [22].

Activation of cardiac fibroblasts can lead to excessive deposition of extracellular matrix (ECM) proteins in the myocardium, leading to cardiac fibrosis [23]. In the context of cardiac diseases, progressive fibrosis is associated with increased collagen content in the myocardium and reduced contractile function [24]. Type I Collagen is a primary constituent of the ECM [25], and *T. cruzi* infection activates mouse cardiac fibroblasts, increasing the expression of ECM [26]. In mammals, collagen genes are responsive to TGFβ1 [27]. It has been previously shown that miR-21 is negatively associated with SPRY1 in the pro-fibrotic pathway [21,28] and enhanced cardiac fibrosis via CADM1/STAT3 pathway in a model of rat cardiac fibroblasts [20]. Our in vitro studies with cardiac fibroblasts showed that miR-21 regulates SPRY1 and CADM1 genes expression, leading to collagens’ upregulation. This study assessed the role of miR-21 for the first time in human cardiac fibroblasts, an important finding with translational perspectives. However, we found no association between miR-21 and *Spry1* or *Cadm1* expression in the hearts of chronic chagasic mice, which could be related to limitations as a single timepoint evaluation and or the methods used in our study. This could also be explained by the involvement of other signaling pathways fibrosis-associated [8,9] regulated by miR-21, which were not explored in this study. Unfortunately, no samples were available to conduct the Western blot experiment to evaluate the expression of a set of proteins.

The in vivo finding of reduced myocardial fibrosis in mice treated with the LNA-miR-21 inhibitor could be explained by a preventive effect (reduced production of ECM components) or by activation of ECM degradation pathways, with the degradation of the scar tissue. Alternatively, both events could occur since fibrosis is present six months post-infection but progresses in the following months as myocardial damage continues to occur [29]. The results from our in vitro studies show that collagen synthesis pathways are suppressed upon short-term miR-21 inhibition, but the in vivo long-term analysis revealed a possible role in the activation of MMP-2 and consequent fibrosis inhibition. Interestingly, we also found increased expression of *Stat3*, a transcriptional regulator of *Mmp2* [30]. Reduced cardiac expression of Mmp2 has been previously associated with cardiac fibrosis, while increased Mmp2 expression contributed to fibrosis resolution in different experimental models [31,32].

In agreement with our previous study [33], we detected *Tgfb1* overexpression in the hearts of chronic chagasic mice, which may influence the Chagas cardiomyopathy’s development by modulation of the immune response [34]. During the acute and chronic phases of Chagas disease, TGFβ1 is involved in the invasion of cardiac fibroblasts and myocytes, intracellular parasite life cycle, regulation of inflammation and immune response, fibrosis, and heart remodeling [35]. TGF-β1 is a pro-fibrogenic factor that regulates fibroblast proliferation, differentiation, apoptosis, and ECM production. We found that TGF-β1 induced the upregulation of miR-21 and collagen genes in mouse and human cardiac fibroblasts. Interestingly, increased *Tgfb1* mRNA expression was also observed after *T. cruzi* infection, which might contribute to miR-21 upregulation in chronic *T. cruzi*-infected mice [14,33,34].

Although our in vitro model is an important system explored to elucidate the mechanisms involved in chronic Chagas cardiomyopathy, it is oversimplified and, therefore, impossible to reproduce all events that occur in the pathological condition. Different molecules involved in the TGF-β1 signaling pathway are mediators of tissue fibrosis [36,37] and could also be involved in the fibrosis reduction observed in our experiments with LNA-antagomir. The interactions between different cell types can amplify the fibrotic response by secretion of growth factors and cytokines amplifying the fibrotic response [38]. Moreover, they are involved in the activation of signaling pathways and transcriptional factors, including mitogen-activated protein kinases (MAPKs), protein kinase B (PKB or AKT), and nuclear factor kappa B (NF-κB) [39,40,41,42,43]. On the other hand, there is a potential paracrine miR-21 crosstalk between cardiac fibroblasts and cardiomyocytes via miRNA-enriched exosomes, leading to cardiomyocyte hypertrophy, which contributes to heart failure [44]. MiR-21 also regulates the activity of extracellular signal-regulated kinases (ERK)/MAPK in cardiac fibroblasts by SPRY1 controlling cardiac hypertrophy [21] that could have contributed with the cardiac fibroblasts. Since the cardiac hypertrophy incidence tends to increase in later timepoints than the one evaluated in our study and the echocardiography showed preserved systolic function even several months after the infection, we not explored the cardiac hypertrophy.

The persistence of cardiac inflammation is a hallmark of chronic Chagas disease. Macrophages present in chagasic heart display high expression of IFN-γ and TNFα [45]. Corroborating with these findings, we found a reduction of IFN-γ after treatment with LNA-anti-miR-21 in macrophages cell culture. Tissue lesions induced by IFN-γ have a direct effect on cardiomyocytes’ gene expression patterns [15]. During the chronic phase of Chagas disease, higher levels of IFN-γ were observed [46], and IFN-γ may play a significant pathogenic role in Chagas disease cardiomyopathy associated with inflammation and fibrosis. Chagas disease cardiomyopathy presented an increased peripheral production of IFN-γ when compared to patients with the asymptomatic and indeterminate form [47]. Consistent with this study, the *IFN-γ* gene expression in the Chagas disease cardiomyopathy experimental model was upregulated compared to naïve controls. A significant reduction in IFN-γ was found in mice treated with LNA-miR-21, which may reduce both inflammation and fibrosis in the hearts of infected mice. In fact, IFN-γ is known to stimulate the production of pro-fibrotic mediators and intracellular pathways involved in the synthesis of collagen and other ECM products in cardiac fibroblasts [48].

The reduction of TNF-α may be beneficial in chronic Chagas disease cardiomyopathy [49]. TNF-α plays an essential role in *T. cruzi* infection associated with cardiac injury [50,51]. Anti-TNF therapy reduced the cardiomyocyte lesion, reinforcing the involvement of this cytokine in promoting cardiac injury in chronic *T. cruzi* infection [52]. Interestingly, our data suggested that the systemic effect of the treatment with LNA-miR-21 inhibitor reduced the serum levels of TNF-α in chronic chagasic mice, which may contribute to fibrosis reduction.

There is still the influence of other miRNAs with altered expression in vivo, especially considering the complexity of the pathogenic mechanisms involved in chronic Chagas disease. In this study, we did not explore other potential miRNAs found dysregulated by PCR array, such as the miR-29 family, which were also previously associated with cardiac hypertrophy and fibrosis [53]. Several factors may modulate miRNAs expression levels and should be further investigated. Further studies are needed to validate miRNAs as biomarkers and therapeutic targets for chronic Chagas disease.

Despite the in vitro target validation and the observation of fibrosis reduction in the heart of chronically *T. cruzi* infected mice, it was not possible to fully elucidate which signaling pathways regulated by miR-21 are involved in the experimental model of Chagas disease. The complexity of the immune response in response to *T. cruzi* infection is associated with a wide variety of proteins regulated by miR-21, resulting in a complex interaction network that is not yet clarified. Unfortunately, it was not possible to explore the immunomodulatory pathways in the context of chronic Chagas disease and still elucidate cytokine targeting by miR-21, even though this would be very interesting to be explored in future studies. Although a limited number of microRNAs were assessed during the screening assays and the small sample size, we demonstrated the involvement of miR-21 and its role as a potential therapeutic target for chronic Chagas disease.

In conclusion, by combining the screening of miRNAs by PCR array, in silico analysis, in vitro validation of miR-21 target genes, and in vivo effect of miR-21 inhibition in the Chagas cardiomyopathy model, we demonstrated for the first time that miR-21 is associated with fibrosis and immune response in mice chronically infected with *T. cruzi*. Our results suggest that targeting miR-21 is a promising therapeutic approach for Chagas cardiomyopathy.

## 4. Materials and Methods

### 4.1. Animals and T. cruzi Infection

Six to eight weeks-old male C57BL/6 mice were used for the experiments. All mice were raised and maintained in the animal facility of the Center for Biotechnology and Cell Therapy, Hospital São Rafael (Salvador, Brazil), and provided with rodent diet and water ad libitum. Animals were handled according to the NIH guidelines for animal experimentation [54]. All procedures described here had prior approval from the local Animal Ethics Committee (CEUA HSR, approval number: 01/2016). Mice were infected by intraperitoneal injection with 1000 *T. cruzi* trypomastigotes from Colombian strain, as previously described [33]. Infection was confirmed by the evaluation of blood parasitemia. After six months of infection, heart samples from chronic chagasic mice (*n* = 8) and naïve controls (*n* = 4) were obtained for the evaluation of the miRNA profile.

### 4.2. Human Samples

The procedures of the study complied with the Declaration of Helsinki and received prior approval from the Ethics Committee of the São Rafael Hospital (approval number: CAAE 20023913.6.0000.0048). Serum samples were obtained from subjects included according to the following criteria: i. Subjects with Chagas disease cardiomyopathy: diagnosis of Chagas disease confirmed by indirect hemagglutination and indirect immunofluorescence; symptomatic heart failure (NYHA classes II, III and IV); left ventricular ejection fraction ≤ 55%, measured by Doppler echocardiogram—Simpson method; the presence of myocardial fibrosis visualized as delayed enhancement in cardiac magnetic resonance imaging (MRI). ii. Subjects with the indeterminate form of Chagas disease: diagnosis of Chagas disease confirmed by indirect hemagglutination and indirect immunofluorescence; absence of clinical diagnosis of heart failure; absence of abnormalities in the echography, Holter and MRI. iii. Healthy controls: the absence of clinical heart failure diagnosis or other known medical conditions, and negative results of a serological test for Chagas disease.

### 4.3. Evaluation of miRNA Expression by RT-qPCR

Total RNA was isolated from mouse heart tissue and human serum samples using the miRCURY RNA isolation kit (Exiqon, Vedbaek, Copenhagen, Denmark). The step of cDNA synthesis was performed using the Universal cDNA Synthesis kit (Exiqon). RNA quantification was performed using NanoDrop™ 1000 (Thermo Fisher Scientific, Waltham, MA, USA). Samples were aliquoted and stored at −80 °C until use. All protocols were performed according to the manufacturer’s recommendations. Real-time PCR assays were performed with miRCURY LNA™ Universal RT microRNA PCR with SYBR^®^ and LNA™ primers set (Exiqon) by 7500 Fast Real-Time PCR System (Life Technologies^®^, Carlsbad, CA, USA). Reference genes for cell (miR-423-3p), tissue (miR-93) and internal positive control (IPC) were tested to normalize the results, according to the manufacturer’s recommendations.

### 4.4. PCR Array

The Pick-and-Mix microRNA PCR Panels custom drawing system and miRCURY LNA™ Universal RT microRNA PCR were used (Exiqon). The array was customized for the detection of 88 microRNAs selected from miRBase (http://www.mirbase.org/, accessed on 2 November 2014), associated with inflammation, fibrosis of cardiac tissue, and arrhythmia. The reference genes and internal positive control (IPC) were placed on each plate to assist in the analysis, and the thermocycler model Applied Biosystems^®^ 7500 Fast Real-Time PCR System (Life Technologies^®^) was used.

### 4.5. In silico Analysis

Microarray profile datasets were composed of heart samples obtained from mice chronically infected with *T. cruzi* (GSE17363), heart transplant biopsies obtained from subjects with Chagas cardiomyopathy (GSE2596), and heart samples obtained from mice in the acute phase of *T. cruzi* infection (GSE41089). Human primary cardiomyocytes (GSE75821) and human primary fibroblasts (GSE13791), both infected with *T. cruzi,* were included in the analyses. We explored microRNA annotation sequences using miRBase (http://www.mirbase.org/, accessed on 7 July 2017) and Ensembl (http://www.ensembl.org/, accessed on 7 July 2017). Enrichment analysis and miRNA-target network were performed to identify miRNA-mRNA target genes using the Ingenuity Pathway Analysis software 2.2.1 (Qiagen, Hilden, Mettmann, Germany). TargetScan (http://www.targetscan.org/, accessed on 7 July 2017) was used to identify target genes based on their respective predicted scores.

### 4.6. In Vitro Studies with Cardiac Fibroblasts

Experiments with human samples were performed after the approval of the institutional Ethics Committee at São Rafael Hospital (approval number: CAAE 20023913.6.000.0048). Human cardiac fibroblasts were isolated from myocardial surgical specimens from patients that underwent cardiac surgery, as described previously [53]. Mouse cardiac fibroblasts were isolated from the hearts of C57BL/6 mice, following a previously described protocol [51]. Cardiac fibroblasts were fixed in PFA 4% for 15 min, washed twice with phosphate-buffered saline (PBS), and incubated overnight at 4 °C with the following primary antibodies: Alpha-SMA (1:100) (Dako, Glostrup, Hovedstaden, Denmark) and Vimentin (1:100) (Santa Cruz Biotechnology, Santa Cruz, CA, USA). On the following day, the cells were incubated for 1 h at room temperature with the following secondary antibodies: anti-mouse IgG Alexa fluor 488-conjugated or anti-goat IgG fluor 488-conjugated both diluted 1:800 (all from Thermo Fisher Scientific). Nuclei were stained with Hoechst 33,342 solution (Thermo Fisher Scientific) diluted 1:1000 in PBS. Image acquisition was performed with the Operetta^®^/Harmony^®^ High Content Screening (HCS) Platform 3.5.2 (PerkinElmer, Waltham, MA, USA). Trypomastigotes of the myotropic *T. cruzi* Colombian strain were obtained from culture supernatants of infected LLC-MK2 cells. Cardiac fibroblasts (10^6^ cells) from mouse and human were incubated with *T. cruzi* (MOI = 10) for 24 h in 6-well plates. For fibroblast activation, 10^6^ cells were incubated with 2 ng/mL recombinant human TGF-β1 (Peprotech, Rocky Hill, NJ, USA) for 24 h at 37 °C incubation and 10% CO_2_ in 6 well plates. Cells were pelleted by centrifugation for RNA or miRNA isolation.

For miR-21 overexpression in human cardiac fibroblasts, 10^6^ cells were nucleofected with 1 µg of the plasmid pcDNA3-miR21 (#21114 Addgene plasmid, Watertown, Massachusetts, USA), P2 solution and program DS-150 in the Nucleofector-4D™ (Lonza, Basel, Switzerland). The cells were incubated at 37 °C for 48 h, dissociated, pelleted by centrifugation, and subjected to mRNA or miRNA isolation. As a control, pMAX-GFP (Lonza) was used to evaluate the transfection efficiency by examination of GFP fluorescence. The efficiency of transfection was estimated to be about 72%, by using a control plasmid encoding GFP.

In vitro experiments were performed in technical and biological triplicates (triplicate wells for each condition in each independent experiment; three experiments performed). The conducted analysis included the mean of technical replicates from each experiment.

### 4.7. Proliferation Assay

Fibroblasts overexpressing miR-21 were stained with 5-Ethynyl-2′-deoxyuridine (EdU) to evaluate the proliferation rate and cell cycle. EdU incorporation into DNA was detected using the Click-iT™ EdU Alexa Fluor^®^ Imaging kit (Thermo Fisher Scientific). All steps of the Click-iT™ reaction were performed at room temperature and protected from light. The cells were fixed using 3.7% formaldehyde for 15 min in phosphate buffered saline (PBS 1x), followed by a 0.5% Triton^®^ X-100 (Sigma Aldrich, St Louis, MO, USA) permeabilization step for 20 min. Click-iT™ EdU reaction cocktail was used according to the manufacturer’s protocol. Nuclei were stained with Hoechst 33,342 solution (all from Thermo Fisher Scientific) diluted 1:2000 in PBS. Images were acquired using the Operetta/Harmony^®^ High Content Screening (HCS) Platform 3.5.2 (PerkinElmer).

### 4.8. In Vitro Inhibition and Blockage of mir-21

MiR-21-LNA-inhibitor and scramble control were designed and purchased from Exiqon. Three different concentrations were tested (10, 25, and 50 nM) and added INTERFERin^®^ (Polyplus, Strasbourg, Illkirch, France), following the manufacturer’s recommendations. Cells were incubated for 24 h.

For the studies of miR-21 blockage with LNA anti-miR-21, mice chronically infected with *T. cruzi* (6 months of infection, *n* = 29) were divided into the following groups: LNA anti-miR-21 (*n* = 8), LNA scrambled (*n* = 8), saline (*n* = 8), and naïve (*n* = 5). Mice received four intraperitoneal (i.*p*.) injections of LNA anti-miR-21 or LNA scrambled (8 mg/kg) diluted in PBS (100 μL) twice a week, with 3–4 day intervals. Controls received an equal volume of vehicle (saline solution). Oligonucleotides were provided by Exiqon. Mice were euthanized two weeks after the last injection (7 months of infection), the hearts were removed and fixed in 10% buffered formalin. Heart sections were performed longitudinally, including atria and ventricles. The percentage of fibrosis in the heart was determined by the analysis of whole sections stained with Sirius red. Semiautomatic morphometric quantification was performed using Image Pro Plus v.7.0 (Media Cybernetics, San Diego, CA, USA), as previously described [55]. Two investigators independently performed the morphometric analyses in a blinded way.

### 4.9. In Vitro Studies with Bone Marrow Derived Macrophage (BMMO)

Bone marrow from C57BL/6 mice were collected by flushing the femurs with RPMI medium (Sigma-Aldrich, St. Louis, MO, USA). The cells were then cultured in 75 cm^2^ flasks at a concentration of 10^6^ cells⁄mL in RPMI medium supplemented with 10% fetal bovine serum (FBS; Gibco Life Technologies, Rockville, MD, USA) and 30% culture supernatant of X-63 cell line (murine myeloma cells which produces GM-CSF), at 37 ° C in a 5% CO2 atmosphere. After 72 h, half of the medium containing non-adherent cells was harvested, and fresh medium were replated. On day 7, BMMO were activated with 1 µg/mL *Escherichia coli* lipopolysacharide (LPS; Sigma-Aldrich) for 24 h. The BMMO were characterized as CD11b, F4/80 double positive cells by three-color flow cytometry. For each sample, data from 100,000 cells were acquired. Fluorescence was measured using a BD LSRFortessa SORP cytometer using BD FacsDiva v.6.2. (Becton Dickinson; Heidelberg, Germany). BMMO were infected with 10^5^ trypomastigotes of Colombian *T. cruzi* strain, obtained from culture supernatants of infected LLC-MK2 cells. Three concentration of LNA-anti-miR-21 (10, 25, or 50 nM) or 25 nM of scramble were added to the BMMO culture. Positive control was 20 µM of benznidazole (an antiparasitic medication). Cell-free supernatants of BMMO were collected 72 h after infection and stocked at −80 °C until used for cytokine measurements. The cells from 24 well-plate were pelleted for PCR analysis and amastigotes were quantified by DRAQ5 Fluorescent Probe (Invitrogen, Rockville, MD, USA) using Operetta High-Content Imaging System (PerkinElmer, Hamburg, DE, Germany). Cells were pelleted by centrifugation and used for RNA extraction. RT-qPCR assays were performed to evaluate the expression levels of *Arg1* (Mm00475988_m1), *Ifnγ* (Mm00801778_m1), and *Il10 (*Mm01288386_m1*)*.

### 4.10. ELISA Assays

Quantification of TNF-α cytokine was performed by ELISA, using a specific antibody kit (RandD Systems, Minneapolis, MN), according to the manufacturer’s instructions. Reaction reading was determined using a spectrophotometer (Spectramax i3, Molecular Devices, San Jose, CA, USA) with 450 nm filter.

### 4.11. RNA Isolation and RT-qPCR Analysis

Heart samples were subjected to total RNA extraction using TRIzol™ reagent (Thermo Fisher Scientific). The RNA concentration was determined by spectrophotometry. Next, cDNA was synthesized, starting with 1 μ*g* RNA using SuperScript™ VILO Master Mix, following the manufacturer’s instructions (Thermo Fisher Scientific). RT-qPCR assays were performed to detect the expression levels of *Tgfb1* (Mm00441724_m1), *Ptprc* (Mm01293577_m1), *Ifn*g (Mm00801778_m1), *Col1a1* (Mm0801666_g1), *Col1a2* (Mm00483921_m1), *Mmp2* (Mm00439498_m1), *Mmp9* (Mm00442991), and *Timp1* (Mm01341361_m1) using TaqMan probes, endogenous control *Hprt* (Mm03024075_m1) and *Gapdh* (Mm99999915_g1), and TaqMan™ Universal PCR Master Mix (all from Thermo Fisher Scientific). *Spry1, Cadm1,* and *Stat3* expression analyses were performed by RT-qPCR assays using SYBR™ Green PCR Master Mix and endogenous control (*Gapdh*). The primer sequences are described in Table 1.

### 4.12. Statistical Analyses

All in vitro experiments were performed in triplicate. The software Exiqon GenEx v6 (Exiqon) and the Threshold Cycle Method [56] were used for RT-qPCR data analysis. The data obtained were evaluated using Student’s *t*-test, ANOVA, and Bonferroni post-hoc test. Graphics were designed by Graphpad Prism v7. A *p*-value of less than 0.05 was considered statistically significant.

## Figures and Tables

**Figure 1 ijms-22-03307-f001:**
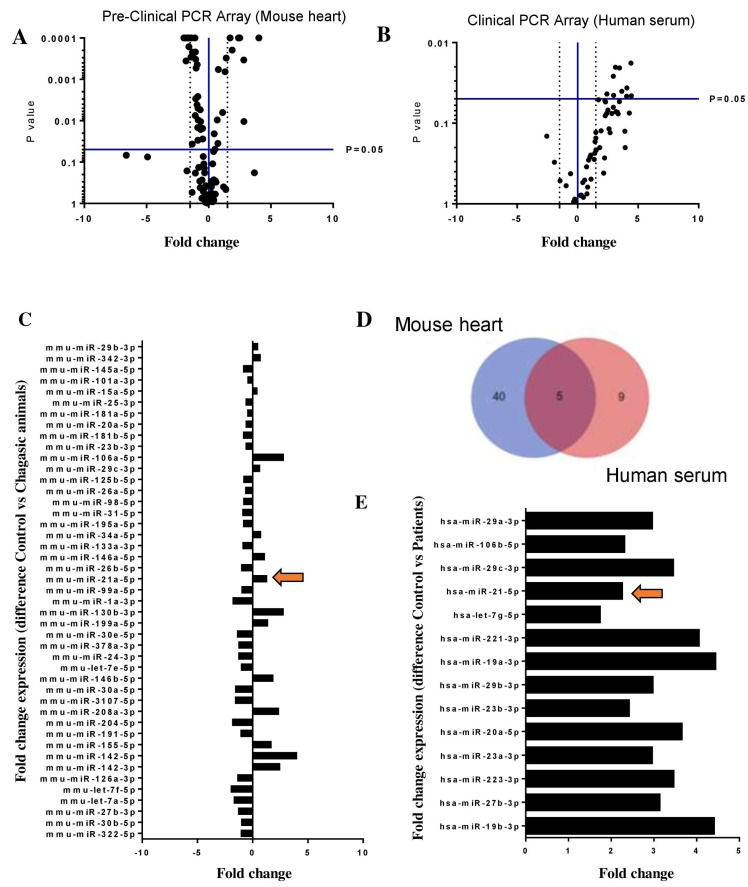
PCR array screening of selected miRNAs in Chagas disease. (**A**) Volcano plot of 88 miRNAs screened by PCR array in heart sample from chronic chagasic (*n* = 8) versus naïve (*n* = 4) mice (**B**) Volcano plot of 88 miRNAs screened by PCR array in serum samples obtained from chronic Chagas disease patients (*n* = 7) compared with healthy subjects (*n* = 3) (**C**) histogram of 45 deregulated miRNAs in heart sample from chronic chagasic (**D**) Venn diagram of PCR array showing five common miRNAs between human serum (pink) and mouse heart (blue) samples (**E**) histogram of 14 upregulated miRNAs in serum samples obtained from chronic Chagas disease patients. GenEX qPCR data analysis (Exiqon), *p* < 0.05; a cutoff value of 1.5-fold change-up and downregulated-was utilized in the analysis and miR-21 upregulation in both arrays is indicated by arrows.

**Figure 2 ijms-22-03307-f002:**
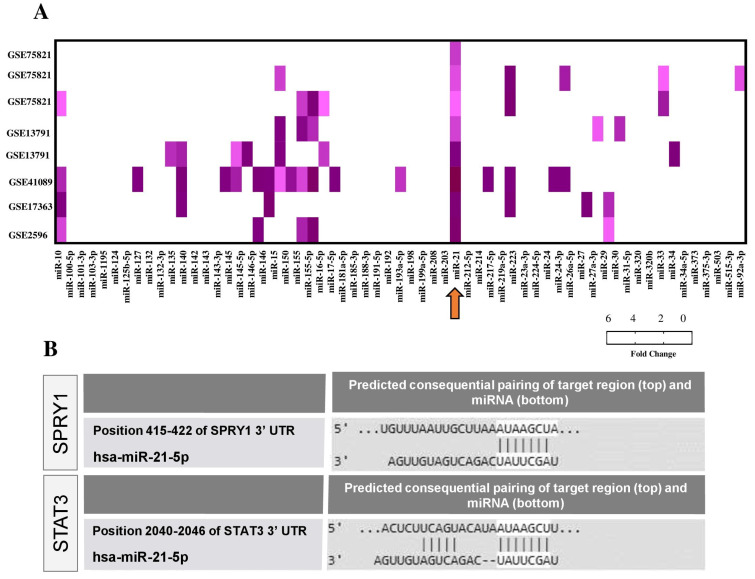
In silico analysis of miRNA-mRNA interactions in Chagas disease. (**A**) miRNA expression in GEO datasets related to Chagas disease presented as -Log10 (*p*-value). Arrow indicates the miR-21 in all datasets. Datasets GSE75821 and GSE13791 referred to the results of in vitro studies that evaluated gene expression in host cells after *T. cruzi* infection. The dataset GSE41089 refers to a study that investigated cardiac mRNA expression during acute *T. cruzi* infection in C57BL/6. GSE17363 includes the results of mRNA expression in heart samples from C57BL/6 mice chronically infected with *T. cruzi.* GSE2596 shows mRNA expression profile in explanted hearts of patients with Chagas cardiomyopathy. The mRNA expression data and *p*-values were exported after being re-analyzed using the GEO2R. (**B**) Predicted consequential pairing of the target region (SPRY1 and STAT3) and conserved sites for miR-21-5p by TargetScanHuman.

**Figure 3 ijms-22-03307-f003:**
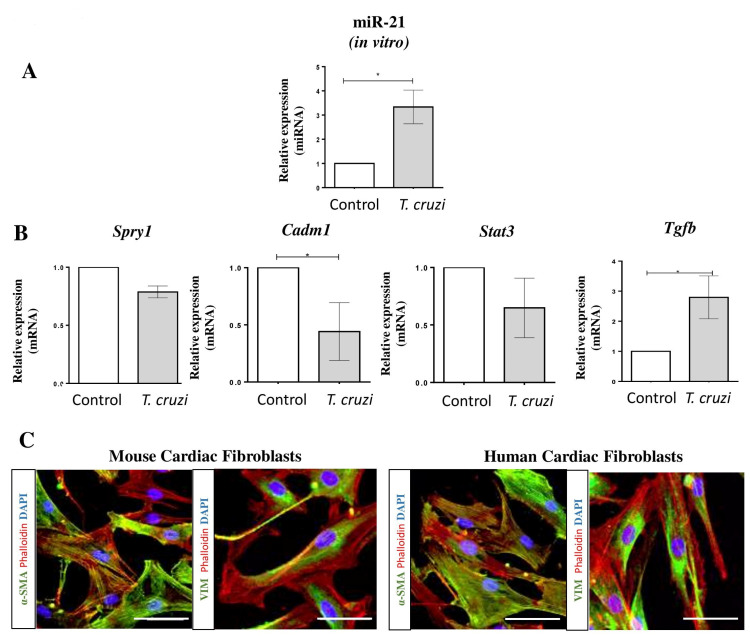
MiR-21 and targets expression after *T. cruzi* infection in mouse cardiac fibroblasts. (**A**) MiR-21 expression in mouse cardiac fibroblasts 24 h after *T. cruzi* infection compared with non-infected control. (**B**) mRNA expression of miR-21 targets Spry1, Cadm1, and Stat3 24 h after *T. cruzi* infection. Tgfb1 mRNA expression was evaluated in vitro after *T. cruzi* infection in mouse fibroblasts. (**C**) Characterization of mouse and human cardiac fibroblasts by immunostaining for alpha-SMA and Vimentin markers (green), phalloidin (red), and nuclei stained with DAPI (blue). Scale bars = 50 µm. Expression analyses were performed by RT-qPCR (fold change to non-stimulated control), miRNA was normalized to miR-423-3p, and mRNA was normalized to HPRT/GAPDH. Data are expressed as mean ± SEM of three independent experiments performed. * *p* < 0.05, Student’s *t*-test.

**Figure 4 ijms-22-03307-f004:**
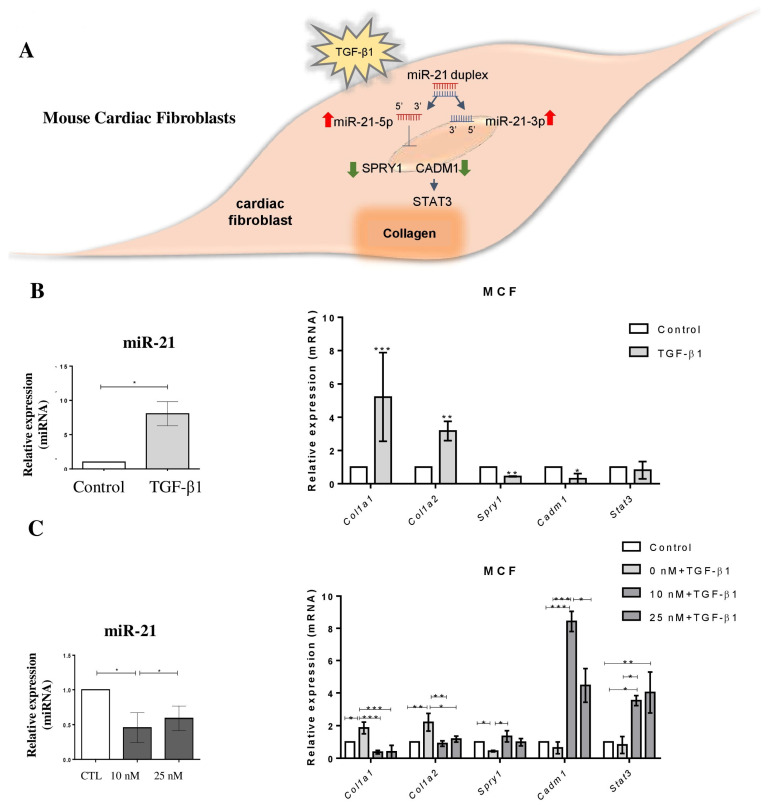
Interplay of TGF-β signaling and miR-21 in mouse cardiac fibroblasts. (**A**) Scheme showing the TGFβ pathway in cardiac fibroblasts. (**B**) Increased expression of miR-*21,*
*Col1a1*, *Col1a2*, *Spry1,* and *Cadm1* in mouse cardiac fibroblasts, 24 h after stimulation with TGF-β1. (**C**) Expression of miR-21 and mRNA expression after stimulation with TGF-β1 and treatment with LNA-anti-miR-21, measured by RT-qPCR (fold change to non-stimulated control). miRNA expression levels were normalized to miR-423-3p, and mRNAs were normalized to *HPRT/GAPDH*. Results are expressed as mean ± SEM of three independent experiments. * *p* < 0.05, ** *p* < 0.01, *** *p* < 0.001, Student’s *t*-test.

**Figure 5 ijms-22-03307-f005:**
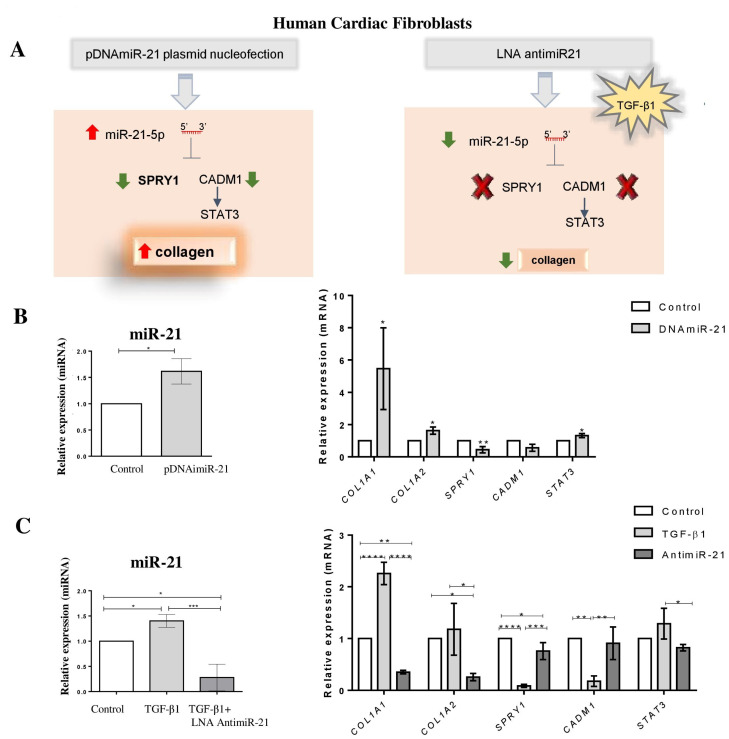
MiR-21 mediates fibrogenic activation in human cardiac fibroblasts. (**A**) Schematic representation of miR-21 overexpression by nucleofection of human cardiac fibroblasts with the expression vector and miR-21 inhibition by LNA anti-miR-21. (**B**) MiR-21, *COL1A1*, *COL1A2, SPRY1*, *CADM1,* and *STAT3* expression levels 48h after nucleofection with miR-21 expression vector (pDNAmiR-21), measured by RT-qPCR (Fold Change to non-stimulated control). miRNA was normalized to miR-423-3p, and mRNAs were normalized to *HPRT/GAPDH*. (**C**) MiR-21, *COL1A1, COL1A2, SPRY1, CADM1,* and *STAT3* expression levels after TGFβ1 stimulation or treated with LNA-anti-miR-21 for 24 h, measured by RT-qPCR (Fold Change to non-stimulated control). miRNA was normalized to miR-423-3p, and mRNAs were normalized to *HPRT/GAPDH*. Results are expressed as mean ± SEM of three independent experiments. * *p* < 0.05, ** *p* < 0.01, *** *p* < 0.001, **** *p* < 0.000.1 Student’s *t*-test.

**Figure 6 ijms-22-03307-f006:**
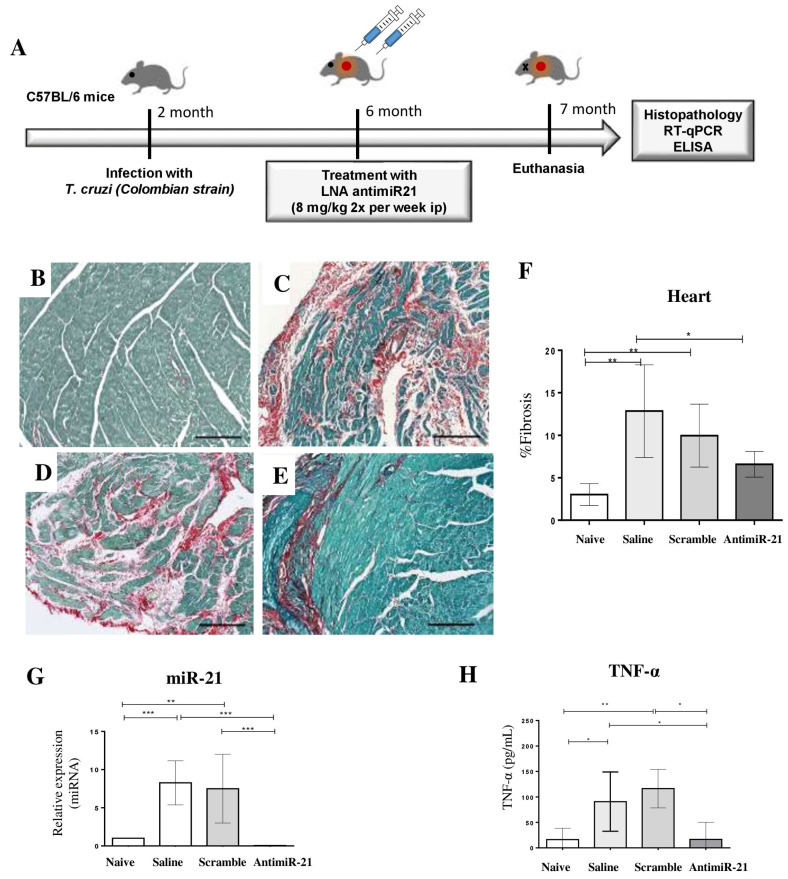
Effects of the treatment with LNA-anti-miR-21 inhibitor in chronic chagasic mice. (**A**) Experimental design of the in vivo experiments. (**B**–**E**) Representative images of cardiac sections (ventricles) stained with Sirius Red. The following experimental groups were evaluated: naïve (**B**), saline (**C**), LNA scrambled (**D**), and LNA-anti-miR-21 (**E**). Scale bars = 100 μm. (**F**) Quantification of fibrosis area. (**G**) Heart samples from mice treated with LNA-miR-21 inhibitor were used to evaluate the miR-21 expression levels by RT-qPCR two weeks after treatment. (**H**) serum levels of cytokine TNF-α evaluated by ELISA. Data are represented as mean ± SEM. *n* = 8 per group. ANOVA with Bonferroni comparisons. * *p* < 0.05. ** *p* < 0.01. *** *p* < 0.001.

**Figure 7 ijms-22-03307-f007:**
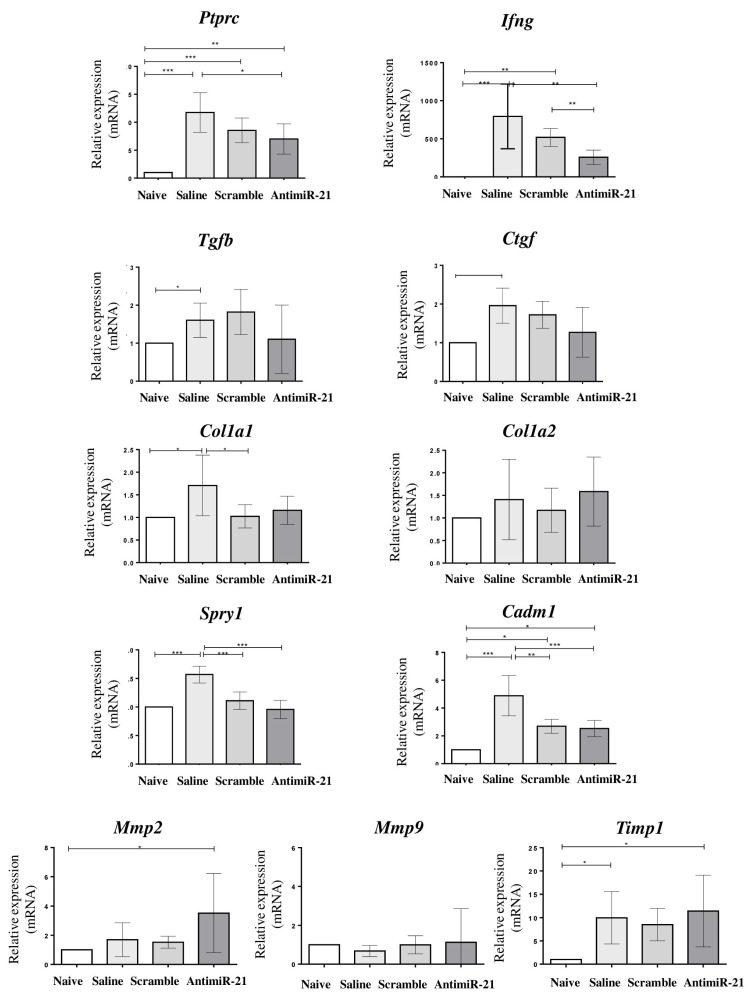
mRNA expression after treatment with LNA miR-21 inhibitor in chronic chagasic mice. Cardiac expression levels of *Ptprc*, *Ifng*, *Tgfb1, Ctgf, Col1a1*, *Col1a2*, *Spry1, Cadm1, Mmp2, Mmp9* and *Timp1* evaluated by RT-qPCR. Data are represented as mean ± SEM. *n* = 8 per group. ANOVA with Bonferroni comparisons. * *p* < 0.05. ** *p* < 0.01. *** *p* < 0.001.

**Table 1 ijms-22-03307-t001:** Primer sequences used in the RT-qPCR assays.

Primers	Forward 5′-3′	Reverse 5′-3′
SPRY1 Mm	ATGGATTCCCCAAGTCAGCAT	CCTGTCATAGTCTAACCTCTGCC
CADM1 Mm	GATCCCCACAGGTGATGGAC	TGATGGTTGCCACTTCTCCTT
STAT3 Mm	CAATACCATTGACCTGCCGAT	GAGCGACTCAAACTGCCCT
GAPDH Mm	GACTCCACTCACGGCAAATTCA	CTGGAAGATGGGCTTC
COL1A1 Hs	GTGCGATGACGTGATCTGTGA	CGGTGGTTTCTTGGTCGGT
COL1A2 Hs	TGGACGCCATGAAGGTTTTCT	TGGGAGCCAGATTGTCATCTC
SPRY1 Hs	GAGAGAGATTCAGCCTACTGCT	GCAGGTCTTTTCACCACCGAA
CADM1 Hs	ATGGCGAGTGTAGTGCTGC	GATCACTGTCACGTCTTTCGT
STAT3 Hs	ACCAGCAGTATAGCCGCTTC	GCCACAATCCGGGCAATCT
GAPDH Hs	GCCAGCATCGCCCCACTTG	GTGAAGGTCAACGGAT
SPRY1 Mm	ATGGATTCCCCAAGTCAGCAT	CCTGTCATAGTCTAACCTCTGCC
CADM1 Mm	GATCCCCACAGGTGATGGAC	TGATGGTTGCCACTTCTCCTT
STAT3 Mm	CAATACCATTGACCTGCCGAT	GAGCGACTCAAACTGCCCT
GAPDH Hs	GACTCCACTCACGGCAAATTCA	CTGGAAGATGGGCTTC

## Data Availability

The data presented in this study are available on request from the corresponding author.

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
