# Peer review of "Therapeutic miR-21 Silencing Reduces Cardiac Fibrosis and Modulates Inflammatory Response in Chronic Chagas Disease"

_ijms, 2021, doi:10.3390/ijms22073307_

Round 1

Reviewer 1 Report

In the current study, the authors detected the elevated expression of miR-21 in the heart of patients and mice infected with T. Cruzi, and in cardiac fibroblasts stimulated by T. cruzi or TGFb. The authors demonstrated that miR-21 directly targeted TGFβ1, SPRY1 and STAT3 and affected the expression of some collagen and ECM remodeling genes through in vitro knockdown and overexpression of miR-21 in mouse cardiac fibroblasts. The author also showed that knockdown of miR-21 in mice reduced cardiac fibrosis caused by T. cruzi infection. Even though the pro-cardiac fibrotic effect of miR-21 was reported a few years ago and has been studied by a number of groups, the research is moderately interesting as it is the first report connecting miR-21 to T. cruzi induced cardiac fibrosis. While the included experiments and data are mostly strong, the manuscript can be further improved through some additional experiments and discussion.

1. Does the expression of miR-21 and Tgfb in cardiomyocytes increase after T. cruzi infection? 

2. The author claimed that Tgfb is a direct target of miR-21. Does miR-21 knockdown increase the expression of Tgfb? If so, how KD of miR-21 lead to improved fibrosis?

Author Response

In the revised version of our manuscript, all of the comments from reviewers were given consideration, and we adapted our text accordingly. Additionally, the manuscript was proofreading and edited by a native English speaker to improve the text even more.

We thank the reviewers for the efforts and time spent in the analysis of our manuscript, which contributed to its improvement. 

Reviewer #1

In the current study, the authors detected the elevated expression of miR-21 in the heart of patients and mice infected with T. Cruzi, and in cardiac fibroblasts stimulated by T. cruzi or TGFb. The authors demonstrated that miR-21 directly targeted TGFβ1, SPRY1 and STAT3 and affected the expression of some collagen and ECM remodeling genes through in vitro knockdown and overexpression of miR-21 in mouse cardiac fibroblasts. The authors also showed that knockdown of miR-21 in mice reduced cardiac fibrosis caused by T. cruzi infection. Even though the pro-cardiac fibrotic effect of miR-21 was reported a few years ago and has been studied by a number of groups, the research is moderately interesting as it is the first report connecting miR-21 to T. cruzi induced cardiac fibrosis. While the included experiments and data are mostly strong, the manuscript can be further improved through some additional experiments and discussion.

  1. Does the expression of miR-21 and Tgfb in cardiomyocytes increase after T. cruzi infection? 

Answer: We thank the referee for the comments. As we focused on fibrosis pathways, we evaluated cardiac fibroblasts, a key cell type involved in this process. However, we included additional literature data in the Discussion section to address the reviewer comment. HiPSC-derived cardiomyocytes were previously utilized in in-vitro studies simulating a cardiac dysfunction microenvironment by stimulation with ET-1 (https://doi.org/10.3390/ijms20164064). In this setting, miR-21 overexpression was demonstrated in cardiomyocytes. Cardiac tissue analysis obtained from patients with chronic Chagas disease also presented increased miR-21 expression (https://doi.org/10.3390/ijms20164064). Moreover, potential paracrine miRNA crosstalk between cardiac fibroblasts and cardiomyocytes via miRNA-enriched exosomes involving miR-21 was also reported (DOI: 10.1172/JCI70577).

  1. The author claimed that Tgfb is a direct target of miR-21. Does miR-21 knockdown increase the expression of Tgfb? If so, how KD of miR-21 lead to improved fibrosis?

Answer: We have revised and modified the manuscript so that our message can be more clearly understood. Although Tgfb was predicted as a miR-21 direct target by in silico analysis (TargetScan), we did not validate this finding with in vitro and in vivo assays. It is clear, though, that miR-21 interfere with the Tgfb pathway according to literature (https://doi.org/10.1159/000479995) (https://doi.org/10.1007/s00395-012-0278-0). As demonstrated in our in vitro assays, the overexpression of miR-21 promotes the pro-fibrogenic activity of Tgfb while knockdown attenuates this activity. In our study, in vivo MiR-21 KD was shown to be associated with reduced cardiac fibrosis, with a tendency towards reduced Tgfb expression in the heart.

Reviewer 2 Report

The present manuscript represents a continuity of a previous study done by the authors in which they have identified miR-21 as a potential biomarker for fibrotic cardiomyopathy associated to Chagas disease. In the current study, the authors selected miR-21 as a candidate therapeutic target and used anti-miR-21 silencing-based approach in vitro and in vivo in order to validate their hypothesis. The paper is of high scientific interest to the readers. However, it contains some flaws that should be addressed:

Point 1: In the first part of the paper, the authors applied several approaches in order to demonstrate the role of miR-21 as a key player in the fibrotic cardiomyopathy associated to Chagas disease. This role is already established either by the authors themselves in their previous studies or by other researchers. Including this part in the manuscript make it redundant. Please comment.

Point 2: Figure 6, Panels b) to e): The authors presented sections of the heart with different treatment (naïve (b), saline (c), 200 LNA scrambled (d), and LNA anti-miR-21 (e)). However, it is well known that most cardiac genes and microRNAs expression are chamber specific and thus, targeting a gene or a microRNA such as miR-21 might not give the same effect in atria and ventricle. Nowhere in the manuscript it is mentioned from which part of the heart are the studied cardiac tissues. Please clarify this point and justify the used strategy. 

Point 3: The authors focalized their study about miR-21 effects in this context on fibrosis and inflammation. Although miR-21 was widely associated to cardiac hypertrophy, it is not mentioned in this study whether miR-21 upregulation in vivo or in vitro lead to any morphological changes in the heart (eg. hypertrophy) or the cardiac fibroblasts (eg. cell size, shape). Please comment. 

 Thank you

Author Response

In the revised version of our manuscript, all of the comments from reviewers were given consideration, and we adapted our text accordingly. Additionally, the manuscript was proofreading and edited by a native English speaker to improve the text even more.

We thank the reviewers for the efforts and time spent in the analysis of our manuscript, which contributed to its improvement. 

Reviewer #2

The present manuscript represents a continuity of a previous study done by the authors in which they have identified miR-21 as a potential biomarker for fibrotic cardiomyopathy associated to Chagas disease. In the current study, the authors selected miR-21 as a candidate therapeutic target and used anti-miR-21 silencing-based approach in vitro and in vivo in order to validate their hypothesis. The paper is of high scientific interest to the readers. However, it contains some flaws that should be addressed:

1: In the first part of the paper, the authors applied several approaches in order to demonstrate the role of miR-21 as a key player in the fibrotic cardiomyopathy associated to Chagas disease. This role is already established either by the authors themselves in their previous studies or by other researchers. Including this part in the manuscript make it redundant. Please comment.

Answer: We thank the reviewer for the comments. We have previously investigated the role of circulating miR-21 as a biomarker for disease severity and also demonstrated increased expression of miR-21 in cardiac tissues of Chagas disease patients (https://doi.org/10.3390/ijms20164064). As a result of these findings, the present work aimed to further investigate if miR-21 could hold potential as a therapeutic target useful for anti-fibrotic therapies. We focused on experiments with cardiac fibroblasts establishing signaling pathways, followed by an in vivo study of a therapeutic miR-21 block in the mouse model of chronic T. cruzi infection, ultimately showing a reduction of fibrosis. The first step was to analyze not only miRNA expression but to find evidence of a molecular signature compatible with miR-21-related gene expression regulation favoring fibrosis, which was not explored in previous studies from the literature. In summary, the data presented herein is a continuation of our previous work and other collaborators, bringing novelty and increasing the knowledge in this field.

2: Figure 6, Panels b) to e): The authors presented sections of the heart with different treatment (naïve (b), saline (c), 200 LNA scrambled (d), and LNA anti-miR-21 (e)). However, it is well known that most cardiac genes and microRNAs expression are chamber specific and thus, targeting a gene or a microRNA such as miR-21 might not give the same effect in atria and ventricle. Nowhere in the manuscript it is mentioned from which part of the heart are the studied cardiac tissues. Please clarify this point and justify the used strategy. 

Answer: We agree that region-specificities may influence the result if sample collection is not performed with a systematic method to ensure comparability and representativeness. We have modified the Methods section to ensure a better description of the procedures used in our study. Sections were performed longitudinally, and the samples included atria and ventricles.

3: The authors focalized their study about miR-21 effects in this context on fibrosis and inflammation. Although miR-21 was widely associated to cardiac hypertrophy, it is not mentioned in this study whether miR-21 upregulation in vivo or in vitro lead to any morphological changes in the heart (eg. hypertrophy) or the cardiac fibroblasts (eg. cell size, shape). Please comment. 

Answer: Although the mouse model utilized in our work display similar histological characteristics to the human disease, cardiac dysfunction does not occur, and the incidence of cardiac hypertrophy tends to increase in later periods than the one evaluated in our study. 

Reviewer 3 Report

The manuscript entitled “Therapeutic miR-21 silencing reduces cardiac fibrosis and modulates the inflammatory response in chronic Chagas disease” is a hypothesis-driven study that aims to explore the therapeutic effect of miR-21 inhibition, a known pro-fibrotic miRNA, in a mouse model of Chagas disease. To answer this question, the authors first screened 88 miRNAs in mouse and human samples, identifying miR-21 as one of the most upregulated miRNAs in Chagas disease. Subsequently, the authors characterize the involvement of miR-21 and cardiac fibrosis using in vitro assays, and finally explores the beneficial effect of miR-21 inhibition in mice infected with T. cruzi. Among the main findings of this study, the authors reported an attenuation in cardiac interstitial fibrosis and inflammation, the Chagas cardiomyopathy hallmarks. The study is current and addresses an interesting topic. Please find some comments below:

Major comments:

  1. All miRNA data were normalized to miR-423 – which is not a standard housekeeping gene. In fact, miR-423 is altered in several heart diseases. Please provide additional data to ensure stable miR-423 expression and suitability for its use as a housekeeping gene.
  2. In this study, all molecular data are based on RNA expression. In addition to that, the authors should also evaluate some protein levels as an extra confirmatory layer.
  3. The involvement of miR-21 in fibroblasts and cardiac fibrosis is widely known. Instead of modulating miR-21 after TGF-b stimulation, the authors should focus on exploring the role of miR-21 in fibroblast function after T. cruzi infection.
  4. The authors claim antimiR-21 therapy alleviates cardiac fibrosis and inflammation in T. cruzi infected mice. Although the findings point towards it, this claim is based only on staining for fibrosis, a TNF-a measurement, and some qPCRs. This section would benefit from both more robust and confirmatory data. Consider generating additional data such as overall survival after therapy, improvements in cardiac function or electrocardiographic data, inflammatory cell infiltration, etc.
  5. A recent study published in a high-impact journal (PMID: 32299591) evaluated the efficacy of antimiR-21 therapy in preventing cardiac dysfunction after I/R injury in pigs with promising results. The beneficial effect of antimiR-21 therapy was credited to reductions in macrophage and fibroblast numbers. The authors should discuss these findings and hypothesize whether the same effect could be happening in this current study. Any attempt to expand the study on this idea would be appreciated.

Minor comments:

  1. Consider adding a Venn diagram comparing both miRNA screenings.
  2. Consider adding the 1.5 FC cut-off in the volcano plots in Figure 1.
  3. The scales in Figure 1 should be resized to accommodate the data better.
  4. Figure 1b – some miRNAs were shown to be upregulated about 10-20 FC in the volcano plot; however, the histogram shows a maximum of 4FC. Please double-check.
  5. Consider merging Figures 1 and 2.
  6. Topic 2.2. – The heading says “TGF-b stimulation,” but there is only data for T. cruzi infected cells.
  7. Consider bringing Figure S1a to Figure 3 to represent T. cruzi-induced fibroblast activation visually.
  8. The sample dispersion (whiskers) is missing in all graphs. Individual values can be generated by calculating the FC for each control against the average control group.
  9. In Figure 2b, if TGF-b is a target of miR-21 (as shown in Figure 2b), why is it upregulated?
  10. Please double-check all the Y-axis labels for errors.
  11. Gene expression data from the same experiment could be represented as a single graph combining all the genes. By doing this, you can free space and merge Figures 4 and 5, which essentially show the same thing but with different cell lines. The same can be applied in Figure 7, which could be merged to Figure 6.

Author Response

In the revised version of our manuscript, all of the comments from reviewers were given consideration, and we adapted our text accordingly. Additionally, the manuscript was proofreading and edited by a native English speaker to improve the text even more.

We thank the reviewers for the efforts and time spent in the analysis of our manuscript, which contributed to its improvement. 

Reviewer #3

Major comments:

  1. All miRNA data were normalized to miR-423 – which is not a standard housekeeping gene. In fact, miR-423 is altered in several heart diseases. Please provide additional data to ensure stable miR-423 expression and suitability for its use as a housekeeping gene.

Answer: We thank the reviewer for the comments. Please notice that not all of the miRNA data were normalized with miR-423, but only the in vitro studies data. This miR showed the best performance, with little to no variation between samples. Therefore, we deemed it suitable as a housekeeping gene in our in vitro experiments. Of note, the same gene has been used as housekeeping by others (https://www.nature.com/articles/s41598-019-49241-7) (https://pubmed.ncbi.nlm.nih.gov/31491899/). We agree that some miRNAs behave differently according to the sample and experimental model. We want to clarify that different candidates were also evaluated, such as U6, miR-16, and miR-93, as recommended by the manufacturer (https://www.qiagen.com/us/products/discovery-and-translational-research/pcr-qpcr-dpcr/qpcr-assays-and-instruments/mirna-qpcr-assay-and-panels/mircury-lna-mirna-focus-pcr-panels/?clear=true#productdetails). Our in vivo data was normalized using MiR-93 as the housekeeping gene, as mentioned in the Method section. Also, we also used a synthetic RNA UniSp6 provide in the miRCURY kit.

  1. In this study, all molecular data are based on RNA expression. In addition to that, the authors should also evaluate some protein levels as an extra confirmatory layer.

Answer: We agree that Western blot analysis could bring useful complementary data. However, all samples were collected to perform histological analysis and PCR assays, while at the same time ensuring enough organ representation for adequate tissue analysis. In order to provide these data, we would need to perform another experiment to obtain heart samples for Western blot, but unfortunately, our laboratory activities have not returned to normal due to the COVID-19 pandemic, and we are unable to repeat the experiments. Also, the conduction of the T. cruzi infection experiments before the sample collection for Western blot has a long-time duration (>8 months). This was included as a study limitation in the discussion section.

  1. The involvement of miR-21 in fibroblasts and cardiac fibrosis is widely known. Instead of modulating miR-21 after TGF-b stimulation, the authors should focus on exploring the role of miR-21 in fibroblast function after T. cruzi infection.

Answer: Our study focused on the chronic phase of Chagas disease, which is characterized by low cardiac parasitism and a strong pro-inflammatory/fibrotic response. Therefore, we focused on modeling the cytokine microenvironment rather than exploring parasite/host interactions and pathways, which would be of great interest, but represent a different research question. Moreover, while miR-21 involvement in cardiac fibrosis of different etiologies has been described, the present study is the first to demonstrate its role in the context of Chagas disease. We also report for the first time the miR-21 involvement after TGF-β stimulation in human cardiac fibroblasts. It is noteworthy to mention that our in vitro experiment was used to validate the targets to be explored in the second step of our experiment, where we tested the effect of miR-21 block in vivo. We agree that in vitro studies in isolated fibroblasts are not ideal to evaluate the complexity of T. cruzi infection, which is why the results of our in vitro studies are complementary to the in vivo analysis.

  1. The authors claim antimiR-21 therapy alleviates cardiac fibrosis and inflammation in T. cruzi infected mice. Although the findings point towards it, this claim is based only on staining for fibrosis, a TNF-a measurement, and some qPCRs. This section would benefit from both more robust and confirmatory data. Consider generating additional data such as overall survival after therapy, improvements in cardiac function or electrocardiographic data, inflammatory cell infiltration, etc.

Answer: Chronic T. cruzi infection in C57Bl/6 mice recapitulates many aspects of the human disease, including histological findings in the myocardial tissue, but fails to induce cardiac dysfunction. We have observed that echocardiography analysis shows preserved systolic function even several months after the infection. We commented on this critical finding in the Discussion section. As we described above (questioning 2), our samples were collected to perform histologic analyses and PCR assays. Unfortunately, the activities at our laboratory were highly impacted by the COVID-19 pandemic, and additional experiments would not be feasible considering their long-term duration (>8 months).

  1. A recent study published in a high-impact journal (PMID: 32299591) evaluated the efficacy of anti-miR-21 therapy in preventing cardiac dysfunction after I/R injury in pigs with promising results. The beneficial effect of anti-miR-21 therapy was credited to reductions in macrophage and fibroblast numbers. The authors should discuss these findings and hypothesize whether the same effect could be happening in this current study. Any attempt to expand the study on this idea would be appreciated.

Answer: It is very exciting to verify the therapeutic potential of miR-21 in different models of chronic heart failure. According to our previous study, we already observed high expression levels of miR-21 in patients with Chagas disease (https://doi.org/10.3390/ijms20164064). Our hypothesis is that the expression of this circulating miRNA may originate in the heart. We also observed miR-21 overexpression in cardiac tissue from patients with chronic Chagas disease. As suggested, we evaluated the effect of antimir-21 on macrophages comparing with benznidazole and Scramble controls. We did not observe a reduction of T. cruzi infection in none of the tested concentrations. However, we observed the interferon gamma reduction and increased levels of Arginine-1 and Interleukin-10. These data were included in our study. We included the study recommended (PMID: 32299591) in our Discussion section.

Figure S2 - (B) The mRNA expression of Arg1, Ifnγ and Il10 in murine macrophages after T. cruzi infection and treated with LNA-anti-miR-21. Data represent the mean ± SEM of three independent experiments, ANOVA, *p<0.05, **p<0.01, ***p<0.001.

Minor comments:

  1. Consider adding a Venn diagram comparing both miRNA screenings.

Answer: Venn diagram was added, as recommended.

  1. Consider adding the 1.5 FC cut-off in the volcano plots in Figure 1. The scales in Figure 1 should be resized to accommodate the data better. Figure 1b – some miRNAs were shown to be upregulated about 10-20 FC in the volcano plot; however, the histogram shows a maximum of 4FC. Please double-check.

Answer: Figure was changed, as suggested.

  1. Consider merging Figures 1 and 2.

Answer: Unfortunately, it was not possible to merge figures 1 and 2. The information presented would be compromised, and difficult to see the details.

  1. Topic 2.2. – The heading says “TGF-b stimulation,” but there is only data for T. cruzi infected cells.

Answer: This sentence was rewritten to make it correct.

  1. Consider bringing Figure S1a to Figure 3 to represent T. cruzi-induced fibroblast activation visually.

Answer: We modified the figures as suggested.

  1. The sample dispersion (whiskers) is missing in all graphs. Individual values can be generated by calculating the FC for each control against the average control group.

Answer: We changed the figure.

  1. In Figure 2b, if TGF-b is a target of miR-21 (as shown in Figure 2b), why is it upregulated?

Answer: We removed the TGFB from in silico analysis (Figure 2) because it was not investigated as a target.

  1. Please double-check all the Y-axis labels for errors.

Answer: We appreciate the input and have corrected the mistakes.

  1. Gene expression data from the same experiment could be represented as a single graph combining all the genes. By doing this, you can free space and merge Figures 4 and 5, which essentially show the same thing but with different cell lines. The same can be applied in Figure 7, which could be merged to Figure 6.

Answer: We modified the figures as much as possible. Distinct assays are demonstrated in Figures 4 and 5, and a merged figure will require the exclusion of some images. The expression of Cadm1 was on a different scale. Then, we decided to maintain figures 6 and 7 as initially presented.

Round 2

Reviewer 2 Report

In this review round, the authors addressed some points raised from the first round. However, some other points remain ambiguous. I'm referring here to point 2 and point 3 from the last review report: 

-I here remind that in point 2, the authors were requested to clarify the tissue samples they have used in their histological/molecular exploration for fibrosis. Although they have mentioned in the revised version that the sections were performed longitudinally including thus atria and ventricle , it is not clear whether fibrosis (histologically) is observed in the whole heart or only in some regions in particular. It is important to add to Figure 6 legend (Panels B,C,D and E ) the heart parts from which the sections are taken (atria or ventricle).

-Answering to point 3 regarding cardiac hypertrophy in miR-21 upregulation model (in vivo and in vitro), the authors mentioned that although the mouse model utilized in their work displays similar histological characteristics to the human disease, cardiac dysfunction does not occur. Although the effect of miR-21 on cardiomyocytes/fibroblasts morphology either in vivo (hypertrophy) or in vitro have been discussed along the manuscript, the author's own findings in this regards haven't been mentioned in any section of the paper which created a real gap. Please add this point to the main manuscript.

-At which stage are the mice models used in this study: neonates? Adults? mixed? please add this detail to the manuscript. 

Author Response

In this review round, the authors addressed some points raised from the first round. However, some other points remain ambiguous. I'm referring here to point 2 and point 3 from the last review report: 

-I here remind that in point 2, the authors were requested to clarify the tissue samples they have used in their histological/molecular exploration for fibrosis. Although they have mentioned in the revised version that the sections were performed longitudinally including thus atria and ventricle, it is not clear whether fibrosis (histologically) is observed in the whole heart or only in some regions in particular. It is important to add to Figure 6 legend (Panels B,C,D and E ) the heart parts from which the sections are taken (atria or ventricle).

Answer: Thank you for this comment. In the mouse model used in our study, fibrosis first appears in the atria and progressively extends to the ventricles over time. Although the quantification was performed in images obtained from both atria and ventricles, all of the representative images included in the figure panel show ventricle sections. We have included this information in the figure caption (Fig 6 line 196), as suggested.

-Answering to point 3 regarding cardiac hypertrophy in miR-21 upregulation model (in vivo and in vitro), the authors mentioned that although the mouse model utilized in their work displays similar histological characteristics to the human disease, cardiac dysfunction does not occur. Although the effect of miR-21 on cardiomyocytes/fibroblasts morphology either in vivo (hypertrophy) or in vitro have been discussed along the manuscript, the author's own findings in this regards haven't been mentioned in any section of the paper which created a real gap. Please add this point to the main manuscript.

Answer: Thank you once again for the careful revision. We have modified the Discussion section (lines 273 and 277) including that miR-21 could be associated with cardiac hypertrophy in vitro by SPRY1, and comments on our own results, as suggested.

-At which stage are the mice models used in this study: neonates? Adults? mixed? please add this detail to the manuscript. 

Answer: All of the experiments were performed in adult mice (6-8 weeks-old at the time of infection, and evaluated after 6 months of infection, during chronic phase), as reported in the Methods section (page 15, lines 315).

Reviewer 3 Report

There are no further comments

Author Response

Thank you once again for the careful revision.